# Effects of Thermomechanical Processing on Hydrogen Embrittlement Properties of UltraHigh-Strength TRIP-Aided Bainitic Ferrite Steels

Tomohiko Hojo [1,*], Yutao Zhou [1,2], Junya Kobayashi [3], Koh-ichi Sugimoto [4], Yoshito Takemoto [5], Akihiko Nagasaka [6], Motomichi Koyama [1], Saya Ajito [1] and Eiji Akiyama [1]

1   Institute for Materials Research, Tohoku University, Sendai 980-8577, Japan; zhou.yutao.s3@dc.tohoku.ac.jp (Y.Z.); motomichi.koyama.c5@tohoku.ac.jp (M.K.); saya.ajito.d1@tohoku.ac.jp (S.A.); eiji.akiyama.d1@tohoku.ac.jp (E.A.)
2   Graduate School of Engineering, Tohoku University, Sendai 980-8579, Japan
3   College of Engineering, Ibaraki University, Hitachi 316-8511, Japan; junya.kobayashi.jkoba@vc.ibaraki.ac.jp
4   School of Science and Technology, Shinshu University, Nagano 380-8553, Japan; sugimot@shinshu-u.ac.jp
5   Graduate School of National Science and Technology, Okayama University, Okayama 700-8530, Japan; tanutake@okayama-u.ac.jp
6   National Institute of Technology, Nagano College, Nagano 381-8550, Japan; nagasaka@nagano-nct.ac.jp
*   Correspondence: tomohiko.hojo.a1@tohoku.ac.jp; Tel.: +81-22-215-2062

**Abstract:** The effects of thermomechanical processing on the microstructure and hydrogen embrittlement properties of ultrahigh-strength, low-alloy, transformation-induced plasticity (TRIP)-aided bainitic ferrite (TBF) steels were investigated to apply to automobile forging parts such as engine and drivetrain parts. The hydrogen embrittlement properties were evaluated by conducting conventional tensile tests after hydrogen charging and constant load four-point bending tests with hydrogen charging. The 0.4 mass%C-TBF steel achieved refinement of the microstructure, improved retained austenite characteristics, and strengthening, owing to thermomechanical processing. This might be attributed to dynamic and static recrystallizations during thermomechanical processing in TBF steels. Moreover, the hydrogen embrittlement resistances were improved by the thermomechanical processing in TBF steels. This might be caused by the refinement of the microstructure, an increase in the stability of the retained austenite, and low hydrogen absorption of the thermomechanically processed TBF steels.

**Keywords:** TRIP-aided bainitic ferrite steel; thermomechanical processing; hydrogen embrittlement; retained austenite





## 1. Introduction

Ultrahigh-strength steel sheets with a tensile strength of 980 MPa and greater have been applied to automobile structural parts to reduce the weights of vehicles and improve collision safety [1]. Thus, in recent years, the mechanical properties and fracture morphologies of high-strength steels were positively investigated [2,3]. In ultrahigh-strength steel sheets, low-alloy transformation-induced plasticity (TRIP)-aided steels [4,5] with a bainitic ferrite matrix (TBF steels) [6] are expected to be the next-generation advanced high-strength steels (AHSS), owing to their high strength associated with a relatively high dislocation density in the matrix and excellent fatigue [7,8] and impact [9,10] properties due to the TRIP effect of retained austenite. The conventional TBF steels were produced by annealing at the austenite region and austempering treatment using cold-rolled steel sheets with chemical compositions of low and medium carbon and an adequate amount of Si and Mn to obtain a microstructure consisting of a bainitic ferrite matrix and retained austenite [6]. On the other hand, downsizing and weight reduction of automobile forging parts such as the engine and drivetrain parts are also required. To resolve these requirements, the TBF

steels are expected to be applied in automobile forging parts, because TBF steels possess the abovementioned excellent properties.

The microstructure evolution of high-strength steels during hot and cold rolling and cooling from an austenitizing temperature were investigated [11,12]. Zhao et al. [13] reported that the refinement and strengthening of advanced high-strength steels were achieved by thermomechanical processing. Sugimoto et al. [14] reported that the refined, recrystallized ferrite in an annealed martensite matrix was obtained when hot forging was conducted on the TRIP-aided annealed martensitic steel at an inter-critical annealing temperature. Moreover, Sugimoto et al. [15–17] reported that strengthening, improved retained austenite characteristics, and improved impact properties were achieved via hot and warm forging of TBF steels. The authors listed in [18,19] reported the effects of hot and warm forging on the microstructure evolution, retained austenite characteristics, and mechanical properties of TRIP-aided martensitic (TM) steels and discussed those mechanisms.

On the other hand, hydrogen embrittlement [20,21], which reduces the ductility of high-strength steels with a tensile strength of more than 980 MPa, also becomes a serious problem, similar to conventional high-strength structural steels. Sojka et al. [22] and Laureys et al. [23] reported that conventional TRIP-aided steels with a tensile strength of 780 MPa were sensitive to hydrogen embrittlement, and fracture morphologies were characterized. The authors [24,25] revealed the following three facts in hydrogen-charged TBF steels. First, the hydrogen embrittlement susceptibility increased with a decreasing strain rate. Second, a plastic strain of 3–10% decreased the hydrogen embrittlement susceptibility. Third, hydrogen-related cracks were initiated at the transformed martensite or bainitic ferrite matrix and transformed martensite interfaces. Furthermore, the effects of the alloying elements on the hydrogen embrittlement of TM steels were investigated, and the addition of alloying elements such as Cr was found to improve the hydrogen embrittlement resistance [26]. However, the hydrogen embrittlement behavior of thermomechanically processed TBF steels has not yet been fully elucidated.

In this study, the hydrogen embrittlement resistance of hot-forged TBF steels was evaluated using a tensile testing technique at a conventional strain rate and a constant load four-point bending technique. The former shows that deformation induced transformation of the retained austenite, unlike the latter. In addition, the effects of thermomechanical processing on the microstructural evolution and hydrogen embrittlement properties of TBF steels were investigated.

## 2. Materials and Methods

Hot-rolled steel bars with a diameter of 32 mm and with chemical compositions as listed in Table 1 were prepared in this study. Hereafter, the steels with carbon contents of 0.2, 0.3, and 0.4 mass% are named steels A, B, and C, respectively. The hot-forging specimens with dimensions of 20 mm in height and 90 mm in length were machined from these steel bars. Hot-forged steels A, B and C were produced via annealing at 930 °C for 1200 s, followed by one-step forging at a 50% reduction ratio ($R$) using a 500-ton press machine (Hydraulic press, Amino Corporation, Fujinomiya, Japan) and subsequent austempering at 350 °C for 1000 s using hot-forging specimens, as depicted in Figure 1. Conventional TBF steels that did not undergo hot forging were produced with $R = 0\%$ via annealing at 930 °C for 1200 s, followed by austempering at 350 °C for 1000 s to compare the effect of thermomechanical processing. Tensile, four-point bending, microstructure observation, and X-ray diffraction specimens were cut from the hot-forged and heat-treated specimens parallel to the rolling direction at a quarter region in the thickness direction of the hot-forged samples.

The microstructure was observed and analyzed using a scanning electron microscope (SEM Merlin, Zeiss, Oberkochen, Germany) equipped with an electron backscatter diffraction (EBSD) system (OIM Data Collection, OIM-Analysis, TSL solutions, Sagamihara, Japan) operated at an accelerated voltage of 20 kV. The EBSD analyses were conducted in an area of 60 μm × 30 μm with a step size of 0.1 μm. The samples for microstructure observation were

ground by waterproof papers of #320 and #600 and were polished using polycrystalline diamond slurries of 9 and 3 μm for the particle size and colloidal silica, respectively.

**Table 1.** Chemical compositions (mass%) of steels.

| Steels | C | Si | Mn | P | S | Al | Nb | Ti | B | O | N |
|--------|------|------|------|--------|--------|-------|-------|------|--------|--------|--------|
| A | 0.20 | 1.52 | 1.50 | 0.004 | 0.0021 | 0.039 | 0.05 | 0.02 | 0.0018 | 0.001 | 0.0011 |
| B | 0.28 | 1.51 | 1.52 | <0.005 | 0.0011 | 0.041 | 0.051 | 0.02 | 0.0016 | 0.001 | 0.0012 |
| C | 0.42 | 1.50 | 1.51 | <0.005 | 0.0009 | 0.043 | 0.05 | 0.02 | 0.0018 | 0.0019 | 0.0035 |

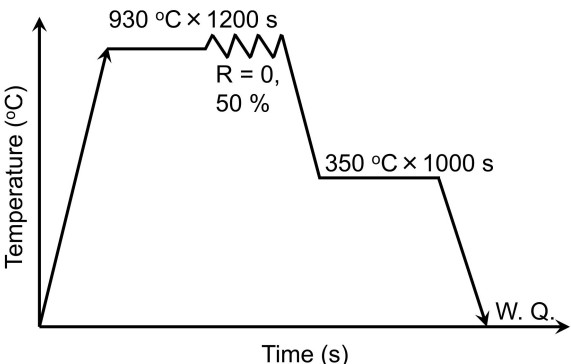

**Figure 1.** Thermomechanical processing and heat treatment diagrams of steels A, B and C. *R* represents reduction ratio. W. Q. indicates water quenching.

The retained austenite characteristics of hot-forged steels A, B, and C were analyzed by the X-ray diffraction (XRD) method using an X-ray diffractometer (Rigaku Co. Ltd., RINT2000, Tokyo, Japan). The volume fraction of the retained austenite ($f_\gamma$) was quantified by the integral intensities of α-Fe(200), α-Fe(211), γ-Fe(200), γ-Fe(220), and γ-Fe(311) diffraction peaks obtained using CuKα radiation. The carbon concentration in the retained austenite ($C_\gamma$) was estimated using Equation (1) [27] from the average lattice parameter ($a_\gamma$ ($\times 10^{-10}$ m)), which was measured from the γ-Fe(200), γ-Fe(220), and γ-Fe(311) diffraction peaks of the CuKα radiation:

$$a_\gamma = 3.5780 + 0.0330C_\gamma + 0.00095Mn_\gamma + 0.0056Al_\gamma + 0.0220N_\gamma + 0.0051Nb_\gamma \\ + 0.0031Mo_\gamma + 0.0039Ti_\gamma, \tag{1}$$

where $Mn_\gamma$, $Al_\gamma$, $N_\gamma$, $Nb_\gamma$, $Mo_\gamma$, and $Ti_\gamma$ represent the concentrations of the respective individual elements (mass%) in the retained austenite. In this study, the contents of the added alloying elements were substituted for these concentrations.

Tensile tests were carried out with a tensile testing machine (AG-X plus 100 kN, Shimadzu Co. Ltd., Kyoto, Japan) at a crosshead speed of 1 mm/min (i.e., an initial strain rate of $8.33 \times 10^{-4}$ /s) at 25 °C with and without hydrogen using tensile specimens with dimensions of 15 mm in gauge length, 6 mm in wide, and 1.2 mm in thickness. The hydrogen embrittlement properties were evaluated by hydrogen embrittlement susceptibility (*HES*), which was calculated using Equation (2) [24,28].

$$HES = (1 - \varepsilon_1 / \varepsilon_0) \times 100\%, \tag{2}$$

where $\varepsilon_0$ and $\varepsilon_1$ denote the total elongation without and with hydrogen, respectively. Hydrogen charging to the tensile specimens was conducted via cathodic charging using a 3 mass% NaCl (Sodium Chloride, FUJIFILM Wako Pure Chemical Co. Ltd., Osaka, Japan) aqueous solution containing 5 g/L NH$_4$SCN (Ammonium Thiocyanate, FUJIFILM Wako Pure Chemical Co. Ltd., Osaka, Japan) at a current density of 10 A/m$^2$ at 25 °C for 48 h before conducting the tensile tests.

Constant load tests were conducted by means of 4-point bending tests using a rectangular specimen with dimensions of 65 mm in length, 10 mm in wide, and 1.2 in mm thickness with hydrogen charging in a 0.5 mol/L $H_2SO_4$ (Sulfuric Acid, FUJIFILM Wako Pure Chemical Co. Ltd., Osaka, Japan) + 0.01 mol/L $NH_4SCN$ (Ammonium Thiocyanate, FUJIFILM Wako Pure Chemical Co. Ltd., Osaka, Japan) solution at 25 °C and a current density of 500 A/m$^2$. The hydrogen embrittlement properties evaluated by the four-point bending tests were defined by the delayed fracture strength (*DFS*), which was the maximum bending stress without failure for 5 h in the specimen.

The hydrogen concentrations in steels A, B, and C were measured by hydrogen thermal desorption analysis (TDA) using gas chromatography. The samples which were charged with hydrogen under the same charging condition as that of the tensile tests were heated between the ambient temperature and 300 °C at a heating rate of 200 °C/h. The diffusible hydrogen concentration was defined as the total hydrogen concentration desorbed between the ambient temperature and 150 °C. After hydrogen charging, the samples were immediately kept in liquid nitrogen to prevent hydrogen desorption. The samples picked up from the cryogenic temperature were rinsed by ultrasonic cleaning using acetone before TDA. The intervals between picking up the sample and the start of TDA were approximately 15 min.

## 3. Results

### 3.1. Microstructure and Tensile Properties

Figure 2 depicts the inverse pole figure (IPF) and phase maps analyzed by EBSD in steels A, B, and C with and without hot forging. In conventional steels A, B, and C without thermomechanical processing, the microstructure of steel A with 0.2 mass%C consisted of a coarse bainitic ferrite matrix and retained austenite, which was located at the packet, block, and lath boundaries. In steels B and C, which exhibited higher carbon contents, the microstructure was characterized as a fine and uniform lath bainitic ferrite matrix and film-type retained austenite located between the packet, block, and lath boundaries. On the other hand, when steels A, B, and C were subjected to thermomechanical processing, the microstructure of steel A was changed to fine ferrite grains with a grain diameter of approximately 10 μm, fine granular retained austenite, and a small amount of bainitic ferrite. In steel B with a carbon content of 0.3 mass%, the fine granular ferrite and bainitic ferrite lath coexisted as a matrix, and fine granular retained austenite was observed at the prior austenitic grain, block, and packet boundaries. In addition, prior austenitic grain refinement and shortened bainitic ferrite laths were achieved, and consequently, a fine and uniform microstructure was achieved by thermomechanical processing in steel C with a carbon content of 0.4 mass%. The film-type retained austenite was located between the shortened bainitic ferrite lath, whereas fine granular retained austenite existed at the fine prior austenitic grain boundaries.

Table 2 lists the retained austenite characteristics and tensile properties of the conventional and hot-forged steels A, B, and C. The initial volume fraction ($f_{\gamma 0}$) and initial carbon concentration ($C_{\gamma 0}$) in the retained austenite of conventional steels A, B, and C were 9.1–17.6 vol% and 0.74–1.22 mass%, respectively, and these increased with the increasing carbon content. The $f_{\gamma 0}$ slightly decreased, and $C_{\gamma 0}$ increased in steel A due to the thermomechanical processing. The $f_{\gamma 0}$ increased and $C_{\gamma 0}$ decreased in steels B and C because of the thermomechanical processing.

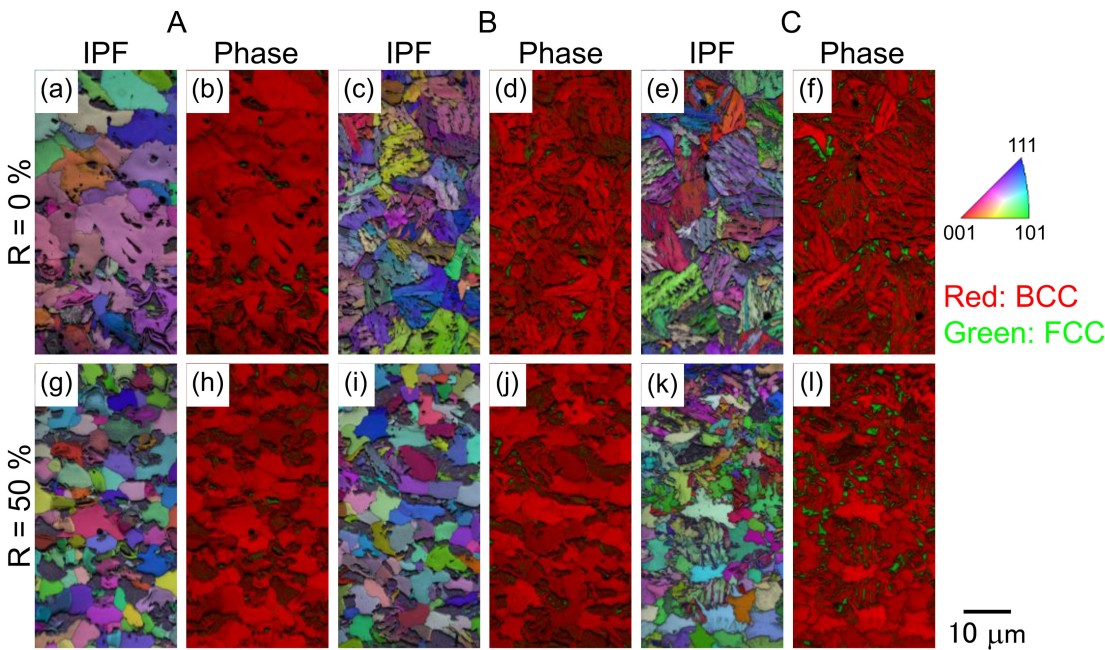

**Figure 2.** (**a,c,e,g,i,k**) Inverse pole figure (IPF) and (**b,d,f,h,j,l**) phase maps of steels (**a,b,g,h**) A, (**c,d,i,j**) B, and (**e,f,k,l**) C (**a–f**) without and (**g–l**) with hot forging. *R* represents reduction ratio. BCC and FCC denote body-centered-cubic and face-centered-cubic, respectively.

**Table 2.** Tensile properties, retained austenite characteristics, and prior austenitic grain sizes of steels A, B and C.

| Steels | R | TS | YS | YR | TEl | UEl | RA | TS × TEl | HES | $f_{\gamma 0}$ | $C_{\gamma 0}$ | d |
|---|---|---|---|---|---|---|---|---|---|---|---|---|
| A | 0 | 748 | 528 | 0.71 | 32.6 | 21.4 | 39.9 | 24.4 | 9.8 | 9.1 | 0.74 | 10.7 |
|   | 50 | 837 | 500 | 0.60 | 25.4 | 21.5 | 36.2 | 21.3 | 26.4 | 8.9 | 0.98 | 7.4 |
| B | 0 | 950 | 749 | 0.79 | 33.2 | 22.5 | 38.7 | 31.5 | 48.2 | 14.1 | 1.09 | 12.8 |
|   | 50 | 959 | 559 | 0.58 | 26.8 | 21.8 | 32.0 | 25.7 | 34.3 | 14.3 | 0.82 | 7.0 |
| C | 0 | 1097 | 937 | 0.86 | 31.9 | 24.4 | 44.7 | 35.0 | 85.3 | 17.6 | 1.22 | 16.3 |
|   | 50 | 1122 | 788 | 0.71 | 27.6 | 24.1 | 32.2 | 31.0 | 68.5 | 21.1 | 1.01 | 7.3 |

*R* (%): reduction ratio, *TS* (MPa): tensile strength, *YS* (MPa): yield strength, *YR*: yield ratio, *TEl* (%): total elongation, *UEl* (%): uniform elongation, *RA* (%): reduction in area, *TS×TEl* (GPa%): strength-ductility balance, *HES* (%): hydrogen embrittlement susceptibility, $f_{\gamma 0}$ (vol%): initial volume fraction of retained austenite, $C_{\gamma 0}$ (mass%): initial carbon concentration in retained austenite, *d* (μm): prior austenitic grain diameter.

Figure 3 depicts the nominal stress–strain curves of the conventional and hot-forged steels A, B, and C without and with hydrogen charging. The tensile properties are shown in Table 2 and Figure 4. The tensile strengths of steels A, B, and C without hydrogen increased from 748 to 837 MPa, from 950 to 959 MPa, and from 1097 to 1122 MPa, respectively, whereas the yield strength decreased from 528 to 500 MPa, from 749 to 559 MPa, and from 937 to 788 MPa, respectively, owing to thermomechanical processing. In addition, thermomechanical processing decreased the total elongation from 32.6 to 25.4%, from 33.2 to 26.8%, and from 31.9 to 27.6%, which corresponded to the reduction ratio in the *TEl* of 22.1%, 19.3%, and 13.5% in steels A, B, and C, respectively, although the uniform elongations of steels A, B, and C without hydrogen were hardly changed via hot forging, and those reduction ratios were −0.5%, 3.1%, and 1.2%, respectively.

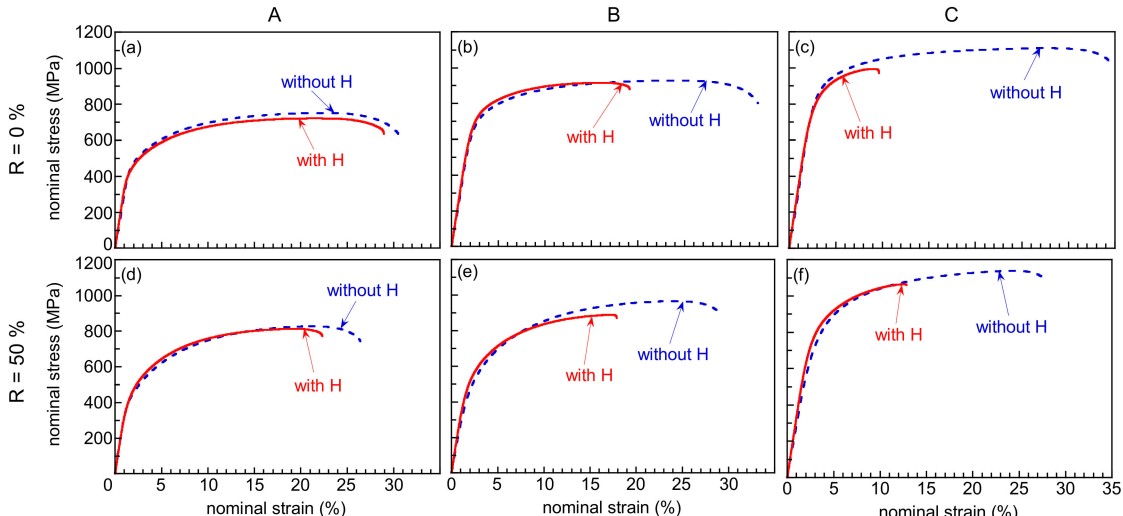

**Figure 3.** Nominal stress–strain curves of (**a**,**b**,**c**) conventional and (**d**,**e**,**f**) hot-forged steels (**a**,**d**) A, (**b**,**e**) B, and (**c**,**f**) C with and without hydrogen. *R* represents reduction ratio.

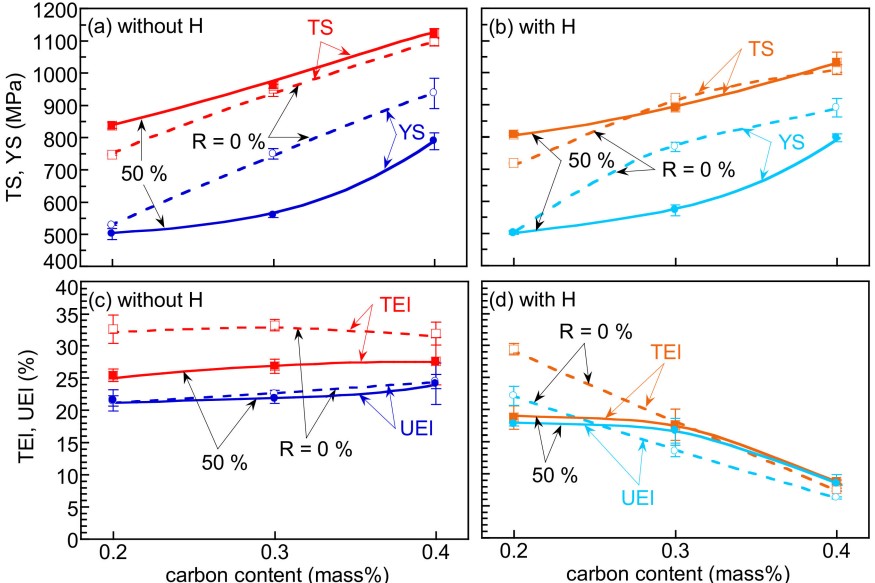

**Figure 4.** Variations in (**a**,**b**) tensile strength (*TS*), yield strength (*YS*), (**c**,**d**) total elongation (*TEl*), and uniform elongation (*UEl*) as a function of carbon content in conventional and hot-forged steels A, B, and C (**a**,**c**) without and (**b**,**d**) with hydrogen. *R* represents reduction ratio.

### 3.2. Hydrogen Embrittlement Properties Evaluated by Tensile Tests

As depicted in Figure 3, steels A, B, and C exhibited a reduction in the total elongation (fracture elongation) due to hydrogen absorption, although the stress–strain response before the fracture hardly changed. The decrease in the total elongation increased with the increasing carbon content in steels A, B, and C. It should be noted that the tendencies of hydrogen-induced mechanical degradation were similar in both the conventional and thermomechanically processed steels A, B, and C. The typical fracture surfaces of the conventional and hot-forged steels A, B, and C with and without hydrogen are depicted in Figure 5. In the steels without thermomechanical processing without hydrogen, dimples with diameters of approximately 1–5 µm appeared on the fracture surfaces of steels A and B, whereas dimples with area fractions of approximately 5.5% and a quasi-cleavage fracture with that of approximately 94.5% coexisted on the typical fracture surface of steel C. When the conventional steels were charged with hydrogen, the fracture surface of steel

A possessed a mixture of dimples which exhibited dimple diameters of approximately 2–10 μm with area fractions of approximately 23.8% and a quasi-cleavage fracture with that of approximately 76.2%, and those of steels B and C were changed to quasi-cleavage fractures. On the other hand, when the steels were subjected to thermomechanical processing, the fracture surface of steel A with hydrogen charging changed from dimples with diameters of approximately 1.5–6 μm to a mixture of dimples with diameters of approximately 2–4 μm and a quasi-cleavage fracture which possessed area fractions of the dimples of 5.3% and quasi-cleavage of 94.7% in the typical fracture surface. On the other hand, steels B and C exhibited fracture surfaces of a mixture of dimples and quasi-cleavage, regardless of the presence of hydrogen.

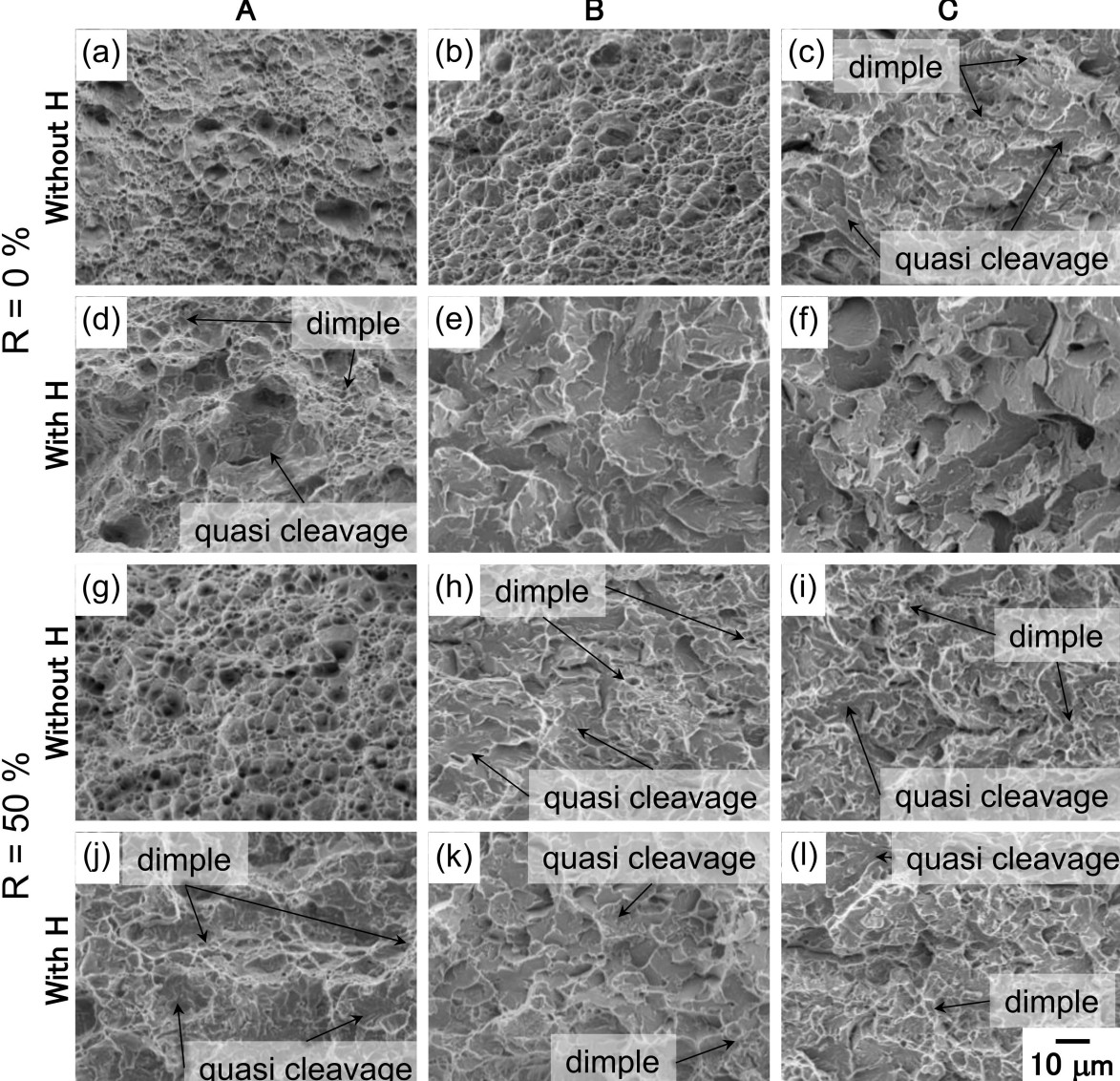

**Figure 5.** Fracture surfaces of (**a–f**) conventional and (**g–l**) hot-forged steels (**a,d,g,j**) A, (**b,e,h,k**) B, and (**c,f,i,l**) C (**d-f,j-l**)with and (**a–c,g-i**) without hydrogen after tensile tests. *R* represents reduction ratio.

Figure 6 depicts the relationships between hydrogen embrittlement susceptibility (*HES*) and tensile strength (*TS*) and yield strength (*YS*) in steels A, B and C. The *HES* increased with increasing *TS* in steels A, B, and C. The thermomechanical processing increased the *HES* of steel A, whereas the *HES* decreased when steels B and C were subjected to hot forging.

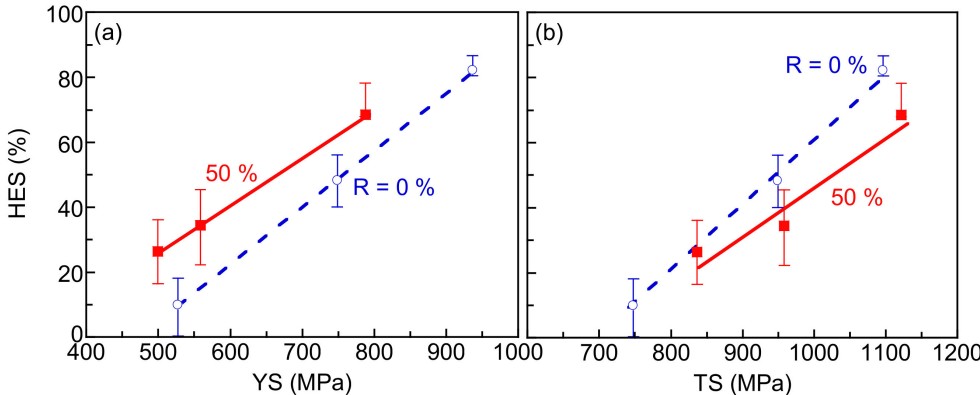

**Figure 6.** Variations in hydrogen embrittlement susceptibility (*HES*) as functions of (**a**) yield strength (*YS*) and (**b**) tensile strength (*TS*) in conventional and hot-forged steels A, B, and C. *R* represents reduction ratio.

Figure 7 depicts the hydrogen desorption curves of the conventional and hot-forged steels A, B, and C. The diffusible hydrogen concentrations obtained from the hydrogen desorption curves in steels A, B, and C are listed in Table 3. It was confirmed that the hydrogen in steels A, B, and C was desorbed between the ambient temperature and approximately 150 °C, and the height of the hydrogen desorption peak increased and its temperature slightly shifted to a higher temperature with an increasing carbon content. The height of the hydrogen desorption peak slightly decreased, and the corresponding diffusible hydrogen concentrations of steels B and C were deduced, although the hydrogen desorption peak temperature did not vary, owing to the thermomechanical processing.

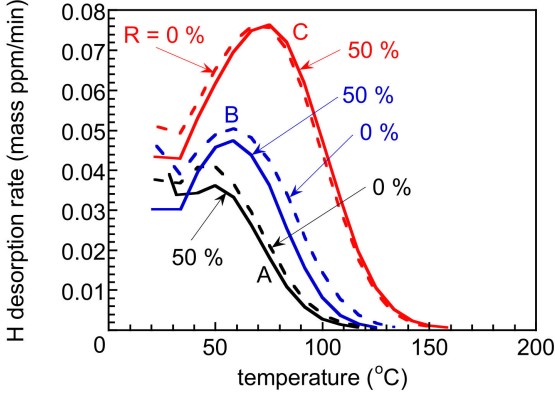

**Figure 7.** Hydrogen desorption curves of conventional and hot-forged steels A, B, and C charged with hydrogen by a 3% NaCl + 5 g/L $NH_4SCN$ solution at a current density of 10 A/m$^2$ for 48 h. *R* represents reduction ratio.

**Table 3.** Hydrogen concentration ($H_C$) of conventional and hot-forged steels A, B, and C. *R* represents reduction ratio.

| Steel | *R* (%) | $H_C$ (Mass ppm) |
|---|---|---|
| A | 0 | 1.36 |
|   | 50 | 1.21 |
| B | 0 | 2.03 |
|   | 50 | 1.64 |
| C | 0 | 3.41 |
|   | 50 | 3.38 |

### 3.3. Hydrogen Embrittlement Properties Evaluated by Four-Point Bending Tests

Figure 8 depicts the applied stress–time to fracture ($\sigma_A$-$t_f$) curves evaluated by the four-point bending tests of the conventional and hot-forged steels A, B, and C. Hydrogen embrittlement did not occur both in steel A with or without thermomechanical processing and in thermomechanically-processed steel B after undergoing four-point bending tests for 5 h. Meanwhile, hydrogen embrittlement occurred in the other steels during the four-point bending tests, and $t_f$ increased with decreasing $\sigma_A$. Figure 9 depicts the relationship between *DFS* and *TS* in the conventional and thermomechanically-processed steels A, B, and C. The *DFS* of steels B and C, which possessed high carbon contents when compared with steel A, improved, owing to the thermomechanical processing. The fracture surfaces of steels B and C after undergoing four-point bending tests are depicted in Figure 10. A quasi-cleavage fracture was observed in the vicinity of the surface of the specimen, where hydrogen embrittlement cracks were initiated in all fractured steels. The fracture surface near the center of the specimen in the thickness direction showed a quasi-cleavage fracture containing flat regions in which facet sizes of 14.1 μm for steel B without thermomechanical processing, 16.4 μm for steel C without thermomechanical processing, and 11.1 μm for steel C with hot forging were similar to the prior austeniti grain or packet sizes of 12.8 μm for steel B without thermomechanical processing, 16.3 μm for steel C without thermomechanical processing, and 7.3 μm for steel C with hot forging, respectively.

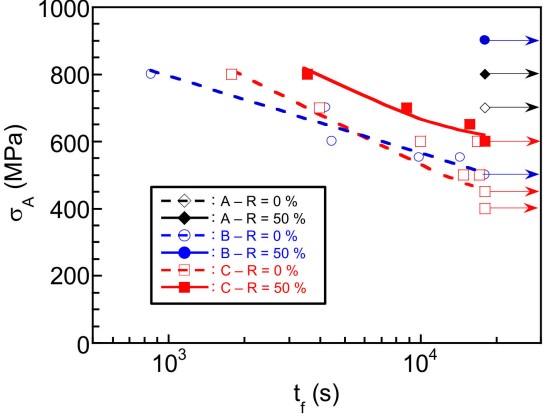

**Figure 8.** Applied stress–time to fracture ($\sigma_A$–$t_f$) curves of conventional and hot-forged steels A, B, and C. *R* represents reduction ratio.

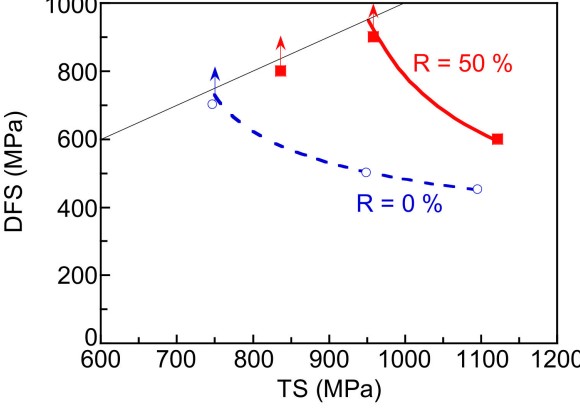

**Figure 9.** Relationship between delayed fracture strength (*DFS*) and tensile strength (*TS*) of conventional and hot-forged steels A, B, and C. *R* represents reduction ratio.

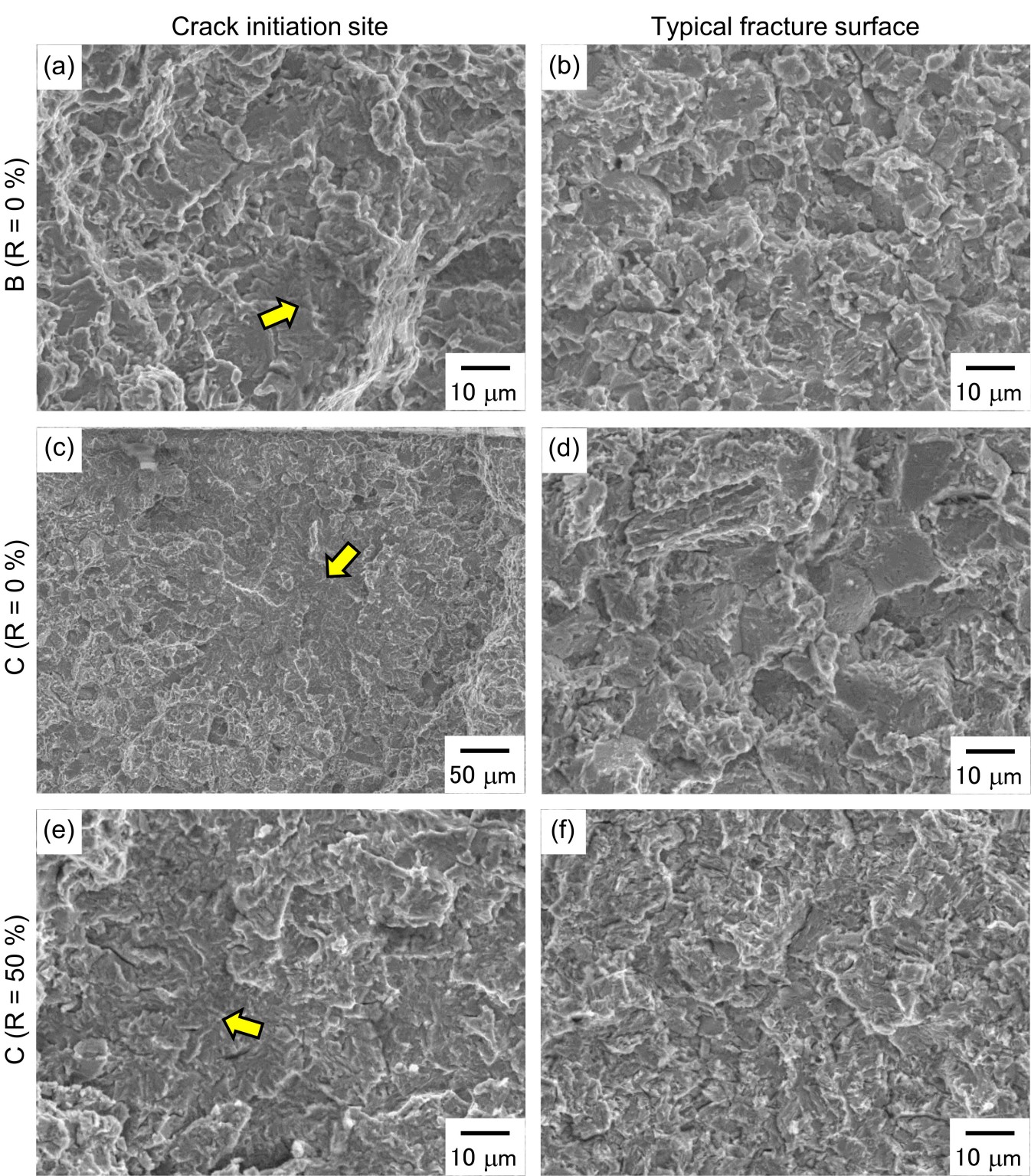

**Figure 10.** Fracture surfaces of (**a**,**b**) conventional steel B and (**c**,**d**) conventional and (**e**,**f**) hot-forged steel C after four-point bending tests, in which arrows represent crack initiation region. *R* represents reduction ratio.

## 4. Discussion

### 4.1. Effects of Thermomechanical Processing on Microstructure Evolution and Mechanical Properties

The microstructure of steel A consisted of fine ferrite grains and a small amount of bainitic ferrite as a matrix and fine granular retained austenite as a second phase, owing

to the thermomechanical processing. The microstructural change during the thermomechanical processing of steel A is illustrated in Figure 11. In steel A without hot forging, austenite grains with a diameter of approximately 15–20 μm were obtained during annealing, and prior austenitic grains remained after the bainite transformation (Figure 11b). When steel A was subjected to thermomechanical processing, fine austenite grains were formed, owing to the dynamic or static recrystallizations at the deformation band in the deformed grain and the piled-up dislocations. A large amount of austenite transformed to ferrite during rapid cooling between the austenite region and austempering temperatures (Figure 11c), and bainite transformation of untransformed austenite might occur during austempering because ferrite and bainite transformations are accelerated when compared with conventional heat treatment without hot forging [29,30] (Figure 11e). As a result, steel A exhibited a fine ferrite matrix, retained austenite, and a small amount of bainitic ferrite (Figure 11d). On the other hand, Figure 12 depicts an illustration of the microstructure evolution mechanism of steel C with thermomechanical processing. In steel C, because the dynamic and static recrystallizations occurred during hot forging, fine austenite grains were newly formed in the deformed austenite in the same way as that in the case of steel A (Figure 12c). However, the transformation to ferrite might have rarely occurred because of the high carbon content of 0.4 mass%, although the ferrite transformation was accelerated. The bainite transformation of a large amount of fine austenite occurred during austempering treatment, and the microstructure of hot-forged steel C was composed of a bainitic ferrite matrix with fine block, packet, and lath structures and a large amount of film-type and granular-type retained austenite (Figure 12d).

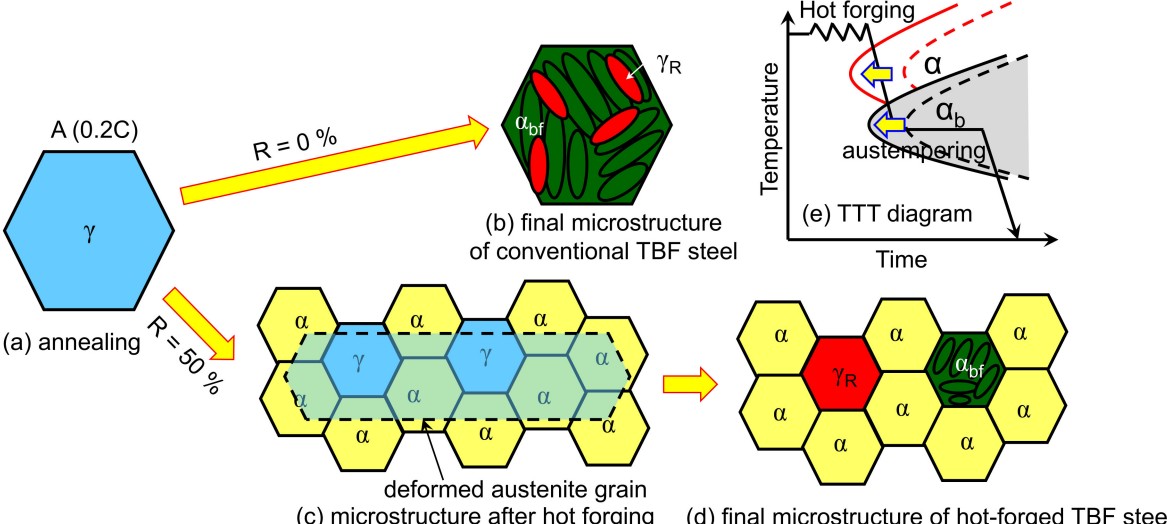

**Figure 11.** Illustrations of microstructure evolution behavior of hot-forged steel A. (**a**) Austenite at annealing temperature, (**b**) final microstructure of conventional TBF steel, (**c**) microstructure after hot forging, (**d**) final microstructure of hot-forged TBF steel, (**e**) TTT diagram, respectively. *R* represents reduction ratio. $\alpha$, $\alpha_b$, $\alpha_{bf}$, $\gamma$, and $\gamma_R$ denote ferrite, bainite, bainitic ferrite, austenite, and retained austenite, respectively. The TTT diagram (**e**) is the time transition temperature diagram.

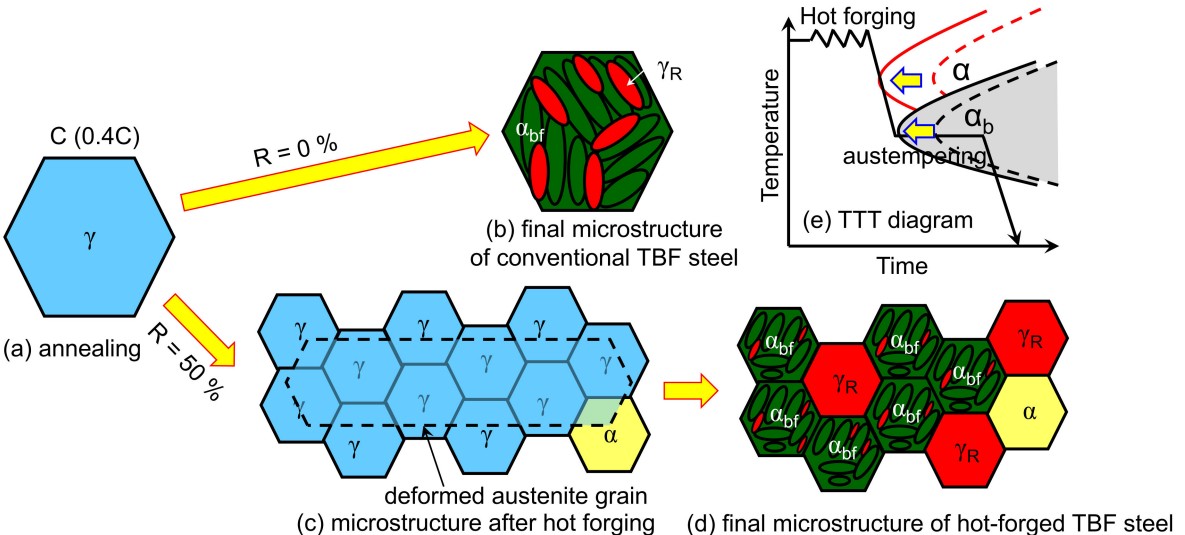

**Figure 12.** Illustrations of microstructure evolution behavior of hot-forged steel C. (**a**) Austenite at annealing temperature, (**b**) final microstructure of conventional TBF steel, (**c**) microstructure after hot forging, (**d**) final microstructure of hot-forged TBF steel, (**e**) TTT diagram, respectively. *R* represents reduction ratio. $\alpha$, $\alpha_b$, $\alpha_{bf}$, $\gamma$, and $\gamma_R$ denote ferrite, bainite, bainitic ferrite, austenite, and retained austenite, respectively. The TTT diagram (**e**) is the time transition temperature diagram.

The yield strength decreased, whereas the tensile strength increased when thermomechanical processing was applied to steels A, B, and C. Moreover, the total elongation decreased, although the uniform elongation hardly changed in steels A, B, and C due to hot forging. As mentioned above, the microstructure of steels A, B, and C possessed fine prior austenitic grain, block, packet, and lath structures, owing to thermomechanical processing. The refinement of the microstructure (i.e., decrease in the grain size) increased the yield strength of the steels according to the Hall–Petch relationship [31,32]. However, it was considered that the yield strength of thermomechanically processed steels A, B, and C decreased because of the formation of ferrite and the premature transformation of block-type retained austenite. In addition, the increase in the tensile strength might have been achieved by the transformation of a large amount of retained austenite at the early stage of plastic deformation, because the volume fraction of the retained austenite increased, and the carbon concentration in the retained austenite decreased, owing to the thermomechanical processing of steels A, B, and C. Moreover, the local deformation capacity might have deteriorated due to the promotion of void initiation at the bainitic ferrite or ferrite matrix and transformed martensite interfaces in the same way as the ferrite-martensite dual-phase steels, because the morphology of the retained austenite varied from filmy to granular, owing to the thermomechanical processing in steels A, B, and C, and the martensite that transformed from block-type retained austenite acted as a hard phase.

### 4.2. Improvement of Hydrogen Embrittlement Properties by Thermomechanical Processing

Steel C showed a decrease in the hydrogen embrittlement susceptibility evaluated by a tensile test and increased the delayed fracture strength evaluated by a four-point bending test, owing to thermomechanical processing, whereas steel A exhibited low hydrogen embrittlement susceptibility and no failure in the four-point bending test, regardless of thermomechanical processing. In general, the hydrogen embrittlement properties of steels A, B, and C with hot forging might be affected by the following factors: variation in the (1) microstructure, (2) characteristics and morphologies of the retained austenite, and (3) hydrogen concentration. For (1) the change in the microstructure owing to thermomechanical processing, it is known that the hydrogen embrittlement properties of high-strength steels are improved by the refinement of the grain size and microstructure.

Zan et al. [33] and Koyama et al. [34] have reported that the hydrogen embrittlement behaviors of the Fe-22Mn-0.6C TWIP steel and the CoCrFeMnNi high-entropy alloy with different grain sizes were investigated, and excellent hydrogen embrittlement resistances were obtained with smaller grain sizes. The grain refinement and the presence of the packet and block as substructures increased the hydrogen embrittlement resistance because of the reduction of the intergranular or quasi-cleavage facet sizes of the fracture surface. Moreover, the refinement of the grain, packet, and block increased the area of the boundaries. When absorbed hydrogen is diffused and trapped at those boundaries, cracking is unlikely to occur because of the decrease in the hydrogen concentration at those boundaries in the steel with fine grains when compared with that with coarse grains. In steels A, B, and C, the refinement of prior austenitic grain, packet, and block sizes was achieved by thermomechanical processing. In particular, steel C possessed a fine bainitic ferrite matrix. Therefore, low hydrogen embrittlement susceptibility and high delayed fracture strength can be achieved by thermomechanical processing. Considering (2) the characteristics and morphologies of retained austenite, it is known that the hydrogen absorption capacity of retained austenite, which is a face-centered cubic (fcc), is higher than that of ferrite with a body-centered cubic (bcc) [35,36]. Thus, hydrogen embrittlement resistance decreased because the cracks were preferentially initiated at the transformed martensite, where hydrogen was supersaturated, and at the bainitic ferrite matrix and transformed martensite interfaces, where hydrogen was diffused and trapped during tensile deformation accompanied with martensitic transformation of the retained austenite. Moreover, it is considered that crack initiation and propagation are accelerated by the premature transformation of retained austenite, which exhibits low stability [37,38]. Thus, the mechanical stability of the retained austenite is considered important for determining the hydrogen embrittlement properties of the steels. It has been reported that the martensitic transformation behavior of retained austenite is affected by the retained austenite morphologies (blocky or filmy types) [39,40] and the surrounding matrix [39–41]. Figure 13 depicts the relationship between the hydrogen embrittlement susceptibility (*HES*) obtained by tensile tests and the retained austenite characteristics in steels A, B, and C. The *HES* tends to increase with an increasing initial volume fraction, initial carbon concentration, and initial total carbon content of the retained austenite. It should be noted that the change in the *HES* might be attributed to the difference in the strength level, resulting from the additive carbon content of the steels. Although the *HES* of steels B and C, which exhibited high volume fractions of retained austenite, were decreased by thermomechanical processing, the effects of the carbon concentration and total carbon content of the retained austenite on the *HES* were small. It was considered that the *HES* of steels B and C with thermomechanical processing was decreased because of the increase in the stability of the retained austenite, owing to its morphology (i.e., fine and film-type retained austenite), the suppression of deformation-induced transformation of the retained austenite, and the corresponding suppressions of crack initiation and propagation, as steels B and C possessed a refined microstructure consisting of fine bainitic ferrite lath structure and fine retained austenite located at the lath boundaries. For (3) the hydrogen concentration, the thermomechanically processed steels A, B, and C possessed a slightly lower hydrogen concentration than conventional steels A, B, and C (Figure 7). It is known that hydrogen absorbed in the steels is trapped at dislocations [42], grain boundaries, lath and packet boundaries [43], and matrix–carbide interfaces [44]. Moreover, austenite of the fcc phase exhibits a high hydrogen absorption capacity when compared with ferrite, bainitic ferrite, and martensite of the bcc phases [35,36]. Figure 14 depicts the relationship between the diffusible hydrogen concentration ($H_C$) and volume fraction of retained austenite ($f_{\gamma 0}$) in steels A, B, and C. The $H_C$ increased with an increase in $f_{\gamma 0}$. However, the $H_C$ of thermomechanically processed steels A, B, and C tended to be lower than those of conventional steels A, B, and C. The increase in the diffusible hydrogen concentration in steels A, B, and C with high additive carbon content is attributed to the increase in the hydrogen trapping sites, such as the retained austenite, which increased with an increasing additive carbon content and the refined bainitic ferrite

matrix. On the other hand, it is considered that the decrease in the $H_C$ in steels A, B, and C with thermomechanical processing was attributed to the decrease in the dislocations in the bainitic ferrite matrix due to the promotion of dynamic and/or static recrystallizations during the hot forging and nucleation of ferrite in the matrix, although the hydrogen trapping sites such as refined prior austenitic grain, packet, and block boundaries and the amount of retained austenite increased. Generally, hydrogen embrittlement may occur because hydrogen is accumulated at the prior austenitic grain, packet, and lath boundaries because of the multiplication and movement of dislocations and the stress-assisted diffusion [45,46] of hydrogen during tensile tests with hydrogen. It can be concluded that the hydrogen embrittlement properties of steels B and C were improved by thermomechanical processing, because the hydrogen concentration at crack initiation sites such as the prior austenitic grain, packet, and block boundaries might not have increased, owing to the refinement of the microstructure.

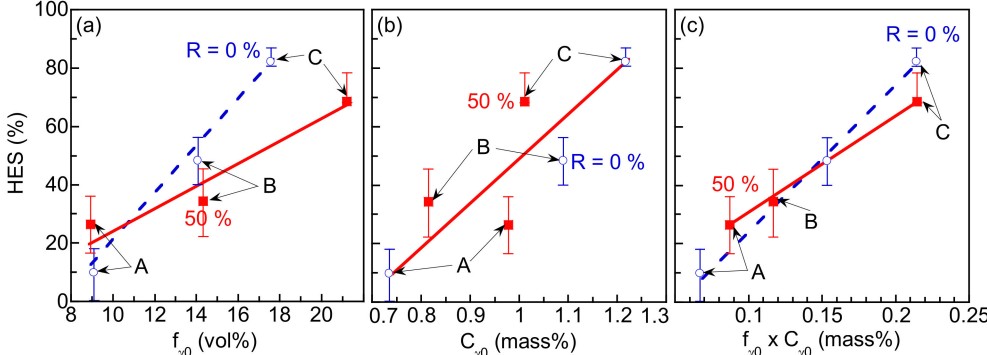

**Figure 13.** Variations in hydrogen embrittlement susceptibility (*HES*) as functions of (**a**) initial volume fraction ($f_{\gamma 0}$), (**b**) initial carbon concentration ($C_{\gamma 0}$), and (**c**) total carbon content ($f_{\gamma 0} \times C_{\gamma 0}$) in retained austenite in steels A, B, and C. *R* represents reduction ratio.

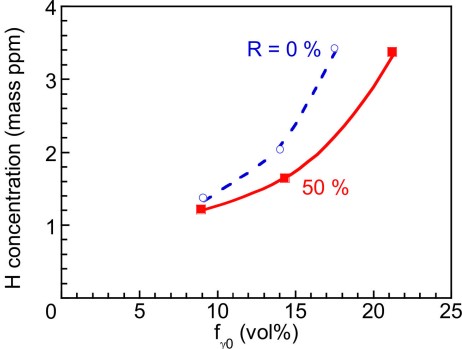

**Figure 14.** Variations in diffusible hydrogen concentration ($H_C$) as a function of initial volume fraction of retained austenite ($f_{\gamma 0}$) in steels A, B, and C. *R* represents reduction ratio.

### 4.3. Evaluation of Hydrogen Embrittlement Properties by Tensile Tests and Four-Point Bending Tests

In this study, the hydrogen embrittlement properties of thermomechanically processed steels A, B, and C were evaluated using the tensile test and four-point bending test methods. In both the tensile and four-point bending tests, it was clarified that the hydrogen embrittlement resistances of steels B and C were improved by thermomechanical processing. In particular, an obvious improvement in the hydrogen embrittlement properties evaluated by the four-point bending tests was obtained, compared with that evaluated by the tensile tests (Figures 6 and 9). As mentioned above, the hydrogen embrittlement resistances of steels A, B, and C might have decreased because a large amount of hydrogen, which was desorbed from transformed martensite, accumulated in the martensite or at the matrix–martensite interfaces, and the crack initiation was accelerated during the tensile

tests. Therefore, obvious high hydrogen embrittlement resistances of steels A, B, and C with thermomechanical processing might not have been obtained in the tensile testing results. On the other hand, in the four-point bending tests, the transformation of retained austenite rarely occurred, owing to the constant loading without plastic deformation and the relatively high stability of the retained austenite, although there was a possibility that the retained austenite with low stability transformed to martensite during the four-point bending tests before hydrogen charging. Consequently, although the hydrogen embrittlement resistance evaluated by the tensile test deteriorated, a significant increase in the delayed fracture strength evaluated by the four-point bending test in thermomechanically processed steels B and C could be achieved. It is considered that in service conditions, the plastic deformation and deformation-induced martensitic transformation of retained austenite do not occur because the constant load or cyclic load below the yield strength are applied to the ultrahigh-strength steels in the automobile forging parts. Therefore, it is suggested that the four-point bending test method is more reasonable for accurately evaluating the hydrogen embrittlement properties than the tensile testing method for steels A, B, and C, which undergo martensitic transformation of the retained austenite accompanied by plastic deformation. However, the hydrogen embrittlement resistances of steels B and C with thermomechanical processing improved regardless of the evaluation methods of both the tensile and four-point bending test methods.

## 5. Conclusions

The effects of thermomechanical processing on the hydrogen embrittlement properties of steels A, B, and C were investigated by means of tensile test and four-point bending constant load test to apply the TBF steels to automobile forging parts. The results are summarized as follows:

(1) The microstructures of steels A, B, and C without thermomechanical processing consisted of a bainitic ferrite matrix and retained austenite. On the other hand, the microstructure of steel A was characterized by a fine polygonal ferrite matrix, a bainitic ferrite, and fine retained austenite, whereas the microstructure of steel C consisted of a refined bainitic ferrite matrix with refined prior austenitic grains, packets, blocks, and laths, and blocky and filmy retained austenite when the steels were subjected to thermomechanical processing. These microstructural changes in steels A, B, and C, owing to thermomechanical processing, were attributed to the promotion of the dynamic and static recrystallizations of austenite grains during hot forging after annealing at the austenite region.

(2) In steels A, B, and C without hydrogen, the tensile strength increased from 748 to 837 MPa, from 950 to 959 MPa, and from 1097 to 1122 MPa, respectively, whereas the yield strength decreased from 528 to 500 MPa, from 749 to 559 MPa, and from 937 to 788 MPa, respectively, by thermomechanical processing. On the other hand, the thermomechanical processing decreased the total elongation from 32.6 to 25.4%, from 33.2 to 26.8%, and from 31.9 to 27.6% in steels A, B, and C, respectively, although the uniform elongation hardly changed from 21.4 to 21.5%, from 22.5 to 21.8%, and from 24.4 to 24.1% in steels A, B, and C, respectively.

(3) The hydrogen embrittlement susceptibility evaluated by tensile tests decreased, and the delayed fracture strength evaluated by the four-point bending tests increased because of thermomechanical processing in steels B and C, whereas the hydrogen embrittlement susceptibility slightly increased in hot-forged steel A.

(4) It is considered that the improvements in the hydrogen embrittlement resistances of steels B and C, owing to thermomechanical processing, were attributed to the suppression of the crack initiations at the prior austenitic grain, bainitic ferrite lath, packet, and block boundaries because of the refined microstructure, improvement of the stability of retained austenite, and decrease in the absorbed diffusible hydrogen concentration.

**Author Contributions:** Conceptualization, T.H., J.K., K.-i.S., and A.N.; methodology, T.H., J.K. and K.-i.S.; validation, T.H., M.K., S.A. and E.A.; formal analysis, T.H.; investigation, T.H., Y.Z., J.K., Y.T. and A.N.; resources, T.H., K.-i.S. and E.A.; data curation, T.H.; writing—original draft preparation, T.H.; writing—review and editing, M.K., S.A. and E.A.; visualization, T.H.; supervision, E.A.; project administration, T.H.; funding acquisition, T.H. All authors have read and agreed to the published version of the manuscript.

**Funding:** This work was supported by a Grant-in-Aid for Scientific Research (C), No. JP18K04743. Moreover, part of this work was financially supported by the Advanced Machining Technology & Development Association and The Amada Fundation, No. AF-2009019.

**Institutional Review Board Statement:** Not applicable.

**Informed Consent Statement:** Not applicable.

**Conflicts of Interest:** The authors declare no conflict of interest. The funders had no role in the design of the study; in the collection, analyses, or interpretation of data; in the writing of the manuscript; or in the decision to publish the results.

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
