# Peer review of "Effects of Thermomechanical Processing on Hydrogen Embrittlement Properties of UltraHigh-Strength TRIP-Aided Bainitic Ferrite Steels"

_metals, doi:10.3390/met12020269_

Round 1

Reviewer 1 Report

Introduction:

"Ultrahigh-strength steel sheets with a tensile strength of 980 MPa and greater have been applied to automobile structural parts to reduce the weight of vehicles and improve collision safety" - needs to add references.

"In ultrahigh-strength steel sheets, low-alloy transformation-induced plasticity (TRIP) [1]-aided steels with bainitic ferrite matrix (TBF steels) [2] associated with TRIP of retained austenite are expected as the next-generation advanced high-strength steels (AHSS) owing to their high strength, excellent fatigue [3,4], and impact [5,6] properties." - the sentence is not clear, try to rewrite.

"On the other hand, downsizing and weight reduction of automobile forging parts such as the engine and drive-train parts are also required. The TBF steels are expected to be applied in automobile forging parts owing to their abovementioned advantages." - again it is not clear what the authors want to say.

Results

Table 2.

Please explain differences in total elongation and uniform elongation, mainly for reduction ratio, 0.

Also, please add plus, minus values, or standard deviation numbers.

Figs 5.

Add to figures arrows with identification of fracture mode. Also, for readers will be good to add some numbers: related to the value of dimples in microns; percentage of fracture mode portions. It is important to value for clear and better identification of differences between the evaluated systems.

Quasi-cleavage fracture is one of the important parameters for the clear identification of Hydrogen Embrittlement.

Part 3.3

"The fracture surface near the center of the specimen in the thickness direction showed a quasi-cleavage fracture containing flat regions in which facet sizes were similar to the prior austenite grain or 236 packet sizes." - this needs more and clear information (numbers), it is an only argument without a scientific background.

Fig. 10 - please, picture are too small for clear readness and identification of investigated crack initiation region.

Discussion.

In this part missing information related to the grain size. Authors needs to add table (?) with numbers.

Conclusions.

First statment, what means "a small amount of bainitic ferrite" - 5% of proportion, 10%, more?

Second statment, authors needs to add explanation about statement related to the uniform and total elongation.

For introduction, add of some following literature:

Kvackaj T, Bidulská J, Bidulský R. Overview of HSS Steel Grades Development and Study of Reheating Condition Effects on Austenite Grain Size Changes. Materials. 2021; 14(8):1988. https://doi.org/10.3390/ma14081988

Heibel, S. Damage mechanisms and mechanical properties of high-strength multiphase steels Materials Volume 11, Issue 59 May 2018 Article number 761 DOI 10.3390/ma11050761

Martínez, C., Briones, F., Villarroel, M., Vera, R. Effect of atmospheric corrosion on the mechanical properties of SAE 1020 structural steel       2018 Materials 11(4), 591

Prislupčák, P.; Kvačkaj, T.; Bidulská, J.; Záhumenský, P.; Homolová, V.; Zimovčák, P. Austenite-Ferrite Transformation Temperatures of C Mn Al HSLA Steel. Acta Metall. Slovaca 2021, 27, 207-209. https://doi.org/10.36547/ams.27.4.1306.

Branco, R., Berto, F. Mechanical behavior of high-strength, low-alloy steels 2018 Metals 8(8),610

Kozłowska, A., Grzegorczyk, B., Morawiec, M., Grajcar, A. Explanation of the PLC effect in advanced high-strength medium-mn steels. A review 2019 Materials 12(24),4175

Author Response

Dear Reviewer 1

Thank you for the valuable comments. Our response is given in attached file. would you please review again our manuscript? 

Best wishes, 

Reviewer 2 Report

The manuscript “Effects of Thermomechanical Processing on Hydrogen Embrittlement Properties of Ultrahigh-Strength TRIP-Aided Bainitic Ferrite Steels” by T. Hojo et al. is trying to assess hydrogen embrittlement susceptibility of TRIP-aided bainitic ferrite steels after thermomechanical processing. The results demonstrated the possibilities of improvement of hydrogen embrittlement resistance of steels by the thermomechanical processing. This may be important to choose the material for the automobile forging parts.

However, it is suggested to be revised before its acceptance for publication. Some corrections are suggested, as follows.

The critical remarks:

  1. The Introduction section does not sufficiently demonstrate the wide range of research conducted on hydrogen embrittlement in steels by scientists from the different countries. Some of the recent references can be added. The authors extensively cite themselves and omitting others works. Self-citations of the authors exceed 50% and should be limited.
  2. Lines 231-232: “The fracture surfaces of steels A, B and C after undergoing four-point bending tests are depicted in Figure 10”. However, there are only the fracture surfaces of steels B and C in Figure 10. The description and caption of Figure 10 should be revised.
  3. Lines 369-370: “It can be concluded that the hydrogen embrittlement properties of steels A, B and C were improved by thermomechanical processing”. This sentence should be revised, since steel A exhibit increasing hydrogen embrittlement susceptibility HES after thermomechanical processing (see data in Table 2 and Figures 6, 13 etc.). The same statement regarding improvement of HES of steel A should be also revised in Lines 385-387 and 409-411.
  4. Lines 421-422: “These microstructural changes in steels A, B and C owing to thermomechanical processing”. What are the changes in steels A, B and C owing to thermomechanical processing? Specify them.
  5. Lines 431-432: “whereas the hydrogen embrittlement properties rarely changed in hot-forged steel A”. This statement should be revised. HES of steel A is increased in ⁓ 2.7 times (from 9.8 to 26.4, see Table 2).

Author Response

Dear Reviewer 2 

Thank you for the valuable comments. Our response is given in attached file. would you please review again our manuscript? 

Best wishes, 

Reviewer 3 Report

The paper compares properties of TRIP steels with different carbon contents and microstructure before and after forging. The steels were characterized in view of the microstructure, mechanical properties, tendency to hydrogen embrittlement and hydrogen content after electrochemical hydrogen charging. The effect of carbon content and retained austenite amount and distribution on the tendency to hydrogen embrittlement is discussed. The paper is well written and contains useful findings that are well explained and supported by experimental evidence or references to earlier studies. Congratulations. I clearly recommend it for publication in Metals.

Minor remarks follow:

  • Line 34-38. For a reader who is not familiar with TRIP steels, this sentence can be hard to understand. I recommend the authors to explain somewhat more in detail what TRIP steel is and what are differences between particular grades.
  • Line 55. Is it true that „plastic strain of 3–10% decreased the hydrogen embrittlement susceptibility“? I believe it is usually the other way around.
  • Line 79-80. Information about a number of replicates is missing. Was each test performed only once? If not, error bars should be added to all charts.
  • Figure 1. All abbreviations should be explained in the figure caption.
  • Line 89. It should read „grinded“ instead of „grounded“.
  • Line 95, 98 and further. Please, consider using a more common notationα-Fe (200) etc.
  • Figure 2. The caption is not comprehensive. It should say at least what is R = 0 % and that FCC = retained austenite.
  • Please, describe exactly the procedure of hydrogen content measurement. How long did it take between the end of hydrogen charging and the analysis? How was the sample treated in between? This is important in order to understand the nature of the analysed hydrogen (freely diffusible or trapped in low energy traps).
  • Figure 8. Axis parameters are not explained in the caption.
  • Figure 11 and 12. Captions should be self-explanatory. Readers do not need to understand what all abbreviations mean.
  • Line 432. Instead of „rarely“, a word „little“ might be more appropriate.

Author Response

Dear Reviewer 3

Thank you for the valuable comments. Our response is given in attached file. would you please review again our manuscript? 

Best wishes, 

Round 2

Reviewer 1 Report

Accept in present form.